# Insights into a Novel and Efficient Microbial Nest System for Treating Pig Farm Wastewater

**DOI:** 10.3390/microorganisms13030685

**Published:** 2025-03-19

**Authors:** Lifei Chen, Lusheng Li, Guiying Wang, Meng Xu, Yizhen Xin, Hanhan Song, Jiale Liu, Jiani Fu, Qi Yang, Qile Tian, Yuxi Wang, Haoyang Sun, Jianqun Lin, Linxu Chen, Jiang Zhang, Jianqiang Lin

**Affiliations:** 1College of Agriculture and Biology, Shandong Province Engineering Research Center of Black Soldier Fly Breeding and Organic Waste Conversion, Liaocheng University, Liaocheng 252000, China; lilusheng@lcu.edu.cn (L.L.); wangguiying@lcu.edu.cn (G.W.); 2210190104@stu.lcu.edu.cn (M.X.); 2210190117@stu.lcu.edu.cn (Y.X.); 17852761158@163.com (H.S.); a19819758188@126.com (J.L.); 15263633726@163.com (J.F.); a2455158524@163.com (Q.Y.); 19063792301@163.com (Q.T.); ozhitiano@outlook.com (Y.W.); 2023405027@stu.lcu.edu.cn (H.S.); 2State Key Laboratory of Microbial Technology, Microbial Technology Institute, Shandong University, Qingdao 266237, China; linxuchen@sdu.edu.cn (L.C.); jianqianglin@sdu.edu.cn (J.L.); 3College of Life Science and Technology, Xinjiang University, Urumqi 830046, China; zhangjiang1@yeah.net

**Keywords:** microbial nest system, piggery slurry, spectral analysis, microbial community, correlation analysis

## Abstract

A microbial nest system (MNS) represents a novel and efficient approach to treating solid–liquid mixtures from pig farming instead of the conventional method, which separates the solid and liquid at first using centrifugation before treating the solid and liquid. However, the key environmental factors influencing the efficiency of this system and the microbial structure are still not clear. This study aimed to elucidate the changes in an MNS considering physicochemical properties, spectral analysis, and correlations between microbial community structures and environmental factors during the treatment. The results showed that the MNS underwent three temperature stages during the treatment process of piggery slurry: a warming period, a high-temperature period, and a cooling period. In the high-temperature period, the most abundant bacterium was *Bacillus*, with a relative abundance of 22.16%, and *Chaetomium* dominated the fungal community with a relative abundance of 11.40%. Moreover, the moisture content, pH value, and electrical conductivity (EC) exhibited an increasing trend, whereas the carbon-to-nitrogen (C/N) ratio and the ratio of ammonia nitrogen to nitrate nitrogen (NH_4_^+^-N/NO_3_^−^-N) showed a decreasing trend. The accumulation of humic acid and fulvic acid suggested that the humification process of organic matter was occurring. The moisture content and C/N ratio were identified as crucial factors influencing the bacterial and fungal community structures, respectively. This study provides a theoretical basis for enhancing the efficiency of piggery slurry treatment using an MNS and rational optimisation of the associated processes.

## 1. Introduction

The demand for the quantity and quality of livestock and poultry meat, eggs, and dairy products is constantly increasing owing to improvements in living standards. Consequently, the number of large-scale breeding farms, which generate a significant amount of animal waste, is increasing. The annual output of livestock and poultry manure is estimated to be approximately 3.8 billion tonnes in China [1,2]. The improper disposal of a solid–liquid mixture of faeces and urine produced by livestock and poultry farms can lead to groundwater pollution and environmental damage [3,4]. A high ammonia nitrogen content in manure can seep into the soil and groundwater and negatively affect water quality [5,6]. Furthermore, the discharge of wastewater with high nitrogen and phosphorus contents can lead to eutrophication, which may result in the death of aquatic organisms and ecological imbalance [4,7]. Farm animal waste is traditionally treated via solid–liquid separation, subjecting the solids to aerobic composting and liquids to anaerobic digestion [8,9,10,11,12]. However, in large-scale water-flushing pig farms, where pig manure is soaked for a long time, it is difficult to achieve solid–liquid separation and the high ammonia nitrogen and high chemical oxygen demand in wastewater make it difficult to achieve harmless treatment and resource utilisation of pig manure in large-scale anaerobic digestion facilities [3,13,14]. Therefore, it is difficult to treat pig wastewater from water-soaked manure farming using traditional methods. Developing a method for efficiently treating pig wastewater from water-soaked manure breeding is currently very urgent in China. The introduction of the microbial nest system (MNS) can directly treat the solid–liquid mixture of piggery slurry without the need for solid–liquid separation. Each cubic metre of the MNS can process 15–20 kg of solid–liquid mixture, effectively solving the waste management problem [15,16]. An MNS not only achieves the resource utilisation of solid faeces but also solves the difficulties in the treatment of liquid wastewater.

An MNS is similar to an ex situ fermentation bed system, both of which evolved from an on-site fermentation bed system [3]. An MNS involves the construction of a microbial fermentation tank outside the livestock house. The tank is lined with padding material, and a specially formulated high-performance microbial agent is added to create a microbial reactor for waste assimilation. The MNS is based on the principle of aerobic biological treatment using fermentation. It adopts the concept of honeycomb as the basic model and utilises agricultural and forestry waste such as sawdust, rice husks, or crop straw as the base padding material [15,16]. When a certain amount of piggery slurry is injected into a reactor, microorganisms decompose carbon–nitrogen-rich macromolecular organic compounds in the manure and gradually convert them into humic acids, ammonium nitrogen, nitrate nitrogen, and other nutrients that are easily absorbed by plants [17,18,19,20]. However, a significant amount of heat is released (at a temperature of up to 80 °C). The entire reactor system maintains a dynamic, cohesive state that resembles a large porous honeycomb structure. Through continuous decomposition in the microbial nest, the solid padding materials in the newly added liquid manure are effectively consumed by microorganisms, and the liquid portion evaporates in a high-temperature evaporation mode, enabling continuous purification of liquid manure from livestock farms [21,22]. Therefore, the MNS is very effective in treating piggery slurry from water-soaked pig farms. Compared with traditional treatment methods, the ability to efficiently treat the solid–liquid mixture of farm manure and water is unique and novel. When the activity of the microbial nest decreases, all nest padding materials become excellent organic fertiliser resources, which can be used for economic crops, seedlings, flowers, and orchards [23]. Compared with traditional composting, the main purpose of the MNS is to process the solid–liquid mixture of the farm because there is not enough land around the farms to absorb the manure in China. The organic fertiliser obtained in the end is its additional product, which reduces the cost of manure treatment. Compared with traditional composting, the MNS needs to inject manure water into the padding material in batches when processing solid–liquid mixture and as the system is affected by factors such as temperature, moisture content, and turning frequency.

Previous studies have mainly focused on the screening of functional microorganisms, such as cellulose-degrading bacteria and thermophilic microorganisms, in the fermentation pile and the characterisation of fermentation characteristics, such as temperature and pH variations in the pile [3,4,9,24]. In China, the MNS has achieved good results in the treatment of piggery slurry in pig farms that do not have enough land to absorb manure and sewage, especially in large-scale water-flushing pig farms. However, the mechanisms underlying manure and sewage treatment in the MNS remain poorly understood. Moreover, there have been no reports on changes in the microbial community structure, including bacterial and fungal diversity, or correlation analyses with the environmental factors in the microbial nest pile. Therefore, in this study, an MNS was used to treat a solid–liquid mixture of piggery slurry. The main research objectives were to (1) determine the physicochemical properties of the fermentation pile during the treatment of piggery slurry using an MNS; (2) determine the spectroscopic characteristics of the fermentation pile material during the treatment of piggery slurry using an MNS; and (3) analyse the microbial community structure in the fermentation pile during the treatment of piggery slurry using an MNS and its correlation with environmental factors. This study could provide theoretical support to improve the efficiency and maintenance of piggery slurry treatment using an MNS.

## 2. Materials and Methods

### 2.1. Experimental Design

The MNS in this study consisted of three fermentation tanks (45 × 6 × 2 m) sourced from the Bocheng Pig Farm near Zhubian Town, Junan County, Linyi City, Shandong Province, China. To construct the MNS model, rice husks were first evenly filled into each fermentation tank, and the laying height was approximately 1.2 m (Figure 1). Next, sawdust was evenly spread on the rice husk, and the laying height was approximately 0.2 m. The volume ratio of rice husk to sawdust was 6:1 (*v*/*v*). Afterwards, a highly efficient compound microbial agent was uniformly sprayed onto the sawdust at a dosage of 0.5% (*w*/*w*). Finally, a self-priming pump was used to spray the piggery slurry onto the padding material, and a tossing machine (Shandong Hongf a Heavy Machinery Co., Ltd., Taian, China) was used to ensure the thorough mixing of all materials until the moisture content reached 40–60% [23]. The MNS was operated for 112 days. During this period, the ratio of piggery slurry to padding material was controlled at 15–20:1 (*w*/*v*); that is, 15–20 kg of piggery slurry was added to each cubic metre of padding material, which was based on the biochemical characteristics of the piggery slurry and the moisture content of the padding material. The tossing machine was operated once a day. The physicochemical characteristics of the raw materials used in the MNS are listed in Table 1. Rice husk, sawdust, and piggery slurry were provided by the Bocheng Pig Farm. The compound microbial agent (*Aspergillus oryzae*: *Saccharomyces cerevisiae*: *Bacillus subtilis* = 1:5:7.5, cfu/cfu) was provided by Shandong Yian Bioengineering Co., Ltd. (Jinan, China) [16].

### 2.2. Sampling

The sampling method involved drawing a line from one end to the diagonal corner of the fermentation tank. Samples were collected at the diagonal endpoints and midpoints of the fermentation pile at a depth of 60 cm. The samples collected from the three points were thoroughly mixed. The sampling time points were the 1st, 3rd, 7th, 28th, 56th, 84th, and 112th days after MNS operation. A portion of the collected samples was stored at 4 °C for physicochemical and spectroscopic analyses. Another portion was stored at −80 °C for subsequent experiments involving microbial diversity and environmental correlation analysis.

### 2.3. Determination of Physicochemical Properties During Piggery Slurry Treatment

The temperature of the fermentation pile at a depth of 60 cm was measured during sample collection. Fresh samples were used to determine the moisture content, pH, electrical conductivity (EC), ammonium nitrogen (NH_4_^+^-N), nitrate nitrogen (NO_3_^−^-N), and the ratio of absorbance values at 465 and 665 nm (E4/E6) using previously described methods [25,26,27]. Dried samples were used for the determination of total carbon and nitrogen using an elemental analyser (Elementar Unicub, Elementar Analysensysteme GmbH, Langenselbold, Germany).

### 2.4. Determination of Spectroscopy Characteristics of Padding Materials from the MNS

A fresh microbial nest of 5 g was gently placed into a 300 mL Erlenmeyer flask containing 50 mL of deionised water. The flask was placed on a horizontal shaker at 150 rpm for 12 h. After shaking, 100 mL of the mixture at 4 °C was obtained and centrifuged at 10,000× *g* for 5 min. The supernatant was then filtered through a 0.45 μm pore size membrane to obtain the extract of the sample. The filtered solution was directly used for fluorescence excitation–emission matrix (EEM) measurements after appropriate dilution using a fluorescence spectrophotometer (Hitachi F-4600, Hitachi, Tokyo, Japan). The parameter settings for the EEM spectroscopy measurements are listed in Appendix A. After appropriate dilution, the filtered solution was freeze-dried using a vacuum freeze-dryer (LD85B3-1; Millrock, Anchorage, AK, USA). The dried samples were then subjected to Fourier transform infrared spectroscopy (FTIR) using a spectrometer (Nicolet iS50, Thermo Fisher Scientific, Waltham, MA, USA). The parameter settings for the infrared spectroscopy measurements are listed in Appendix A.

### 2.5. DNA Extraction and High-Throughput Sequencing

Sampling was conducted at various time points during the microbial nest padding fermentation process: day 1 (labelled A), day 3 (labelled B), day 7 (labelled C), day 28 (labelled D), day 56 (labelled E), day 84 (labelled F), and day 112 (labelled G). Three replicate samples were collected at each time point and labelled as A1, A2, A3; B1, B2, B3; C1, C2, C3; D1, D2, D3; E1, E2, E3; F1, F2, F3; and G1, G2, and G3. Total genomic DNA from frozen padding samples was extracted using the FastDNA Spin Kit for Soil (MP Biomedicals, Irvine, CA, USA) according to the manufacturer’s instructions. The extracted genomic DNA was analysed using 1% agarose gel electrophoresis.

PCR amplification was performed on the bacterial 16S rRNA gene V3-V4 and the fungal internal transcribed spacer gene ITS1-ITS2 hypervariable region at different time points of the microbial nest padding samples. The primer sequences selected were 338F-806R, 338F (5′-ACTCCTACGGGAGGCAGCAG-3′), and 806R (5′-GGACTACHVGGGTWTCTAAT-3′), and were used to amplify the bacterial V3-V4 region of 16S rRNA genes. The primers ITS1 (5′-CTTGGTCATTTAGAGGAAGTAA-3′) and ITS2 (5′-TGCGTTCTTCATCGATGC-3′) were used to amplify the fungus ITS1-ITS2 region of ITS1 genes. The amplification conditions were as follows: initial denaturation at 95 °C for 3 min; 27 cycles of denaturation at 95 °C for 30 s; annealing at 55 °C for 30 s; an extension at 72 °C for 45 s; and a final extension at 72 °C for 10 min. The PCR products were analysed using 2% agarose gel electrophoresis. Subsequently, gel extraction of the PCR products was performed using an Omega Gel Extraction Kit (Omega Bio-Tek, Norcross, GA, USA). After elution with Tris-HCl, purified products were subjected to 2% agarose gel electrophoresis. Based on the preliminary quantification results of electrophoresis, PCR products were quantified using a NanoDrop 2000 nucleic acid spectrophotometer (Thermo Fisher Scientific, Waltham, MA, USA). The PCR products were mixed in appropriate proportions according to the sequencing requirements of each sample and then submitted to Majorbio BioPharm Technology Co., Ltd. (Shanghai, China) for sequencing on an Illumina MiSeq platform (Illumina Inc., San Diego, CA, USA). The bacterial and fungal sequencing data were uploaded to the NCBI Sequence Read Archive (SRA) database with accession numbers PRJNA778917 and PRJNA778923, respectively.

### 2.6. Data Analysis

Raw sequencing data were denoised using exact sequence variants [28]. Chimeric sequences were removed using USEARCH based on the SILVA database [29]. Operational taxonomic units (OTUs) were determined using BLAST 2.12.0 by searching for representative sequences against the SILVA database v132 using the ‘best hit’ approach [23]. The community richness index (Chao1) and community diversity indices (Simpson and Shannon) used to estimate α-diversity were calculated using QIIME 2 and visualised using R software v3.2.0 (https://cran.r-project.org/src/base/R-3/ (accessed on 23 November 2022)). The relative abundance of the order is displayed in a stacked column. For β-diversity analysis, principal coordinate analysis (PcoA) based on a weighted UniFrac distance was used to visualise the OTU data. Taxonomic cladograms were constructed based on the OTUs. Spearman’s correlation coefficients (r < −0.6, *p* < 0.01) between probiotics and ammonifiers were calculated via R using the psych package and were used to construct co-occurrence network analyses. The relationship between bacterial and fungal communities and environmental parameters was analysed using redundancy analysis (RDA) in Canoco for Windows (Version 4.5). SPSS software (version 22.0; SPSS Inc., Chicago, IL, USA) was used for the comprehensive and systematic statistical analyses of all monitored test data.

## 3. Results and Discussion

### 3.1. Physicochemical Properties of Compost in the MNS

Temperature is a critical parameter that reflects the maturity of microbial nests during composting. The fermentation process of the entire microbial nest pile could be divided into three stages (Figure 2a): the warming stage (1–7 days), the high-temperature stage (7–84 days), and the cooling stage (84–112 days). This temperature trend was similar to that observed previously [10,20] and could be the result of a variety of microbial metabolic activities at different stages. During the warming stage, bacteria and fungi primarily metabolised soluble small-molecule organic compounds, including sugars and proteins. The degradation of organic matter releases a large amount of heat that causes microbial nest piles to show continuous increases in temperature [3,4]. Owing to the uneven distribution of the pile and its poor thermal conductivity, the generated heat is not easily dissipated, and it could result in a rapid increase in temperature within a short period [30]. When the temperature reached 55 °C, the pile entered the high-temperature stage. In the high-temperature stage, some actinobacteria, fungi, and heat-resistant Bacillus spp. exhibited vigorous metabolism and possibly decomposed a large amount of organic matter in the pile, including recalcitrant macromolecular organic compounds such as cellulose, hemicellulose, and lipids. Additionally, the continuous turning of the pile during the high-temperature stage leads to a significant evaporation of moisture, reducing the moisture content in the pile [31,32]. Therefore, it is necessary to continuously add piggery slurry to the piles to remove manure from the water-soaked faeces of pig farms [33]. As the easily degradable organic matter in the microbial nest pile is depleted, the microbial metabolic activity in the pile reduces, leading to a gradual reduction in the heat released by microorganisms and a continuous decrease in the pile temperature [23].

Water is essential for the growth and metabolic activity of microorganisms in microbial nests. Soluble organic compounds in nests must dissolve in water before they can be absorbed and utilised by microbial cells [34]. At the initial stage, padding, such as sawdust and rice husks, accumulates and spreads in the fermentation tank with a moisture content of approximately 20%, which is not conducive to microbial reproduction and metabolism (Figure 2b). Therefore, on the first day, the piggery slurry was sprayed into the microbial nest to maintain the moisture content (at 40–60%) in the entire fermentation tank, which was necessary during the heating and high-temperature periods from day 1 to day 84 of the fermentation cycle. This is because, during the early and middle stages of fermentation, microbial metabolism in the nest is vigorous and releases a significant amount of heat. With the continuous turning of the turning machine, a large amount of water evaporated; this observation is consistent with the findings of [3]. However, when the paddings in the microbial nests, such as sawdust and rice husks, were completely decomposed or tended to decay, the organic matter content in the nest became insufficient to support microbial metabolism. Consequently, the heat released from the nest decreased, and a large amount of water could not be discharged. Therefore, after 84 days, during the late stage of fermentation, the moisture content in the microbial nest increased, and the MNS’s efficiency in treating manure and sewage decreased [27,35].

During the initial stage of 1–7 days, the pH in the microbial nest increased from 6.48 to 6.97 (Figure 2c), contradicting the observations in traditional composting. In traditional composting, the anaerobic fermentation stage leads to an increase in acidic substances, such as organic acids, which causes a decrease in pH [10,36]. However, during the microbial nest process, it is possible that the decomposition of manure produces ammonium ions, which neutralise the accumulated organic acids and, thus, increase the pH [27]. With the continuous addition of piggery slurry and turning of the microbial nest during the accumulation and fermentation process from day 7 to day 84, microbial metabolism was active, and organic compounds such as sugars, proteins, and lipids were continuously decomposed. In addition, the ammonia nitrogen content continued to increase, and the pH was between 7.0 and 7.62, indicating an alkaline trend. After 84 days, the pH sharply increased to 8.66, indicating that the continued addition of the piggery slurry burdened the microbial nest. The degradation rate of organic matter was reduced at this stage, and the production of small organic acids that inhibit the release of ammonia nitrogen was reduced, leading to an increase in pH [3,9].

EC can reflect the changes in soluble salts during microbial nest fermentation. The EC in the microbial nest varies with the degradation of organic matter and changes based on the contents of organic acid salts, ammonium salts, and phosphate salts [37]. EC showed a continuously increasing trend (Figure 2d). During the microbial nest fermentation process, based on the data shown in Figure 2d, the content of EC is positively correlated with the concentration of soluble salts to a certain extent, as statistically analysed [38]. In the initial stage, from day 1 to day 28, EC increased from 426.3 μs/cm to 646.7 μs/cm and remained between 646.7 μs/cm and 678.3 μs/cm from day 28 to day 84, with minimal overall changes. This may be because microbial growth and metabolism are vigorous during mid-term fermentation, and organic salt substances are decomposed and utilised by microorganisms, with the utilised substances transforming into the structure of microbial cells. Additionally, ammonia nitrogen, which is converted into ammonia gas and released at a high temperature, decreases the ammonium salt content [39]. Consequently, the EC was maintained within a certain range. In the late stage of fermentation, the EC of the microbial nests rapidly increased from 678.3 μs/cm to 864.7 μs/cm; this could be attributed to the delay in the decomposition of added manure in the later stages, resulting in the accumulation of large amounts of inorganic and organic salts, which indirectly reflected the increase in moisture content. During this period, the growth and metabolism of microorganisms are inhibited, and microbial diversity is compromised.

Changes in carbon and nitrogen are important characteristics that reflect variations in microbial nest fermentation processes. The C/N ratio is a crucial indicator of compost fermentation and maturation [40]. The content changes in TN and TC are shown in Appendix A. During the initial stage, the microbial nests mainly consisted of sawdust and rice husks, which contained large amounts of lignin, cellulose, and hemicellulose (Figure 2e). At this stage, the total carbon content exceeded the total nitrogen content needed for microorganisms. The nitrogen source was only replenished after the addition of the piggery slurry to the microbial nest. Therefore, the C/N ratio during the initial stage was approximately 40–50. As the fermentation process in the microbial nest progressed, many microorganisms proliferated, grew, and metabolised, decreasing the C/N ratio continuously. The C/N ratio in the late stage of fermentation on day 112 was only 16.62, which is lower than the optimal value of 25–30 required to sustain microbial growth and metabolism [5,41]. Therefore, in the later stages, the microbial nest tended to mature with stagnant growth and metabolism. These results are consistent with those of previous studies [4,9].

The ratio of NH_4_^+^-N to NO_3_^−^-N is an evaluation index for compost maturity, and the determination of ammonium nitrogen and nitrate nitrogen is essential for compost samples [42]. The ratio of NH_4_^+^-N to NO_3_^−^-N is an important indicator of compost maturity. When the ratio of NH_4_^+^-N to NO_3_^−^-N is less than one, the compost samples tend to mature [43]. The content changes of NH_4_^+^-N and NO_3_^−^-N are shown in Appendix A. The fermentation process in a microbial nest is also referred to as nitrification. During the initial stage, the ratio of NH_4_^+^-N to NO_3_^−^-N increased from 14.3 to 15.2 due to an increase in the ammonia nitrogen concentration (Figure 2f). As the microbial nest composting process entered the high-temperature phase, the ratio of NH_4_^+^-N to NO_3_^−^-N rapidly decreased from 15.2 to 0.38 on the 112th day. On the 84th day, the ratio of NH_4_^+^-N to NO_3_^−^-N was 0.64, which met the standard for compost maturity of <1. Therefore, the compost samples were considered mature from the 84th day of microbial nest composting.

Humic acid substances exhibit specific absorption peaks at 465 and 665 nm, and the ratio of absorbance values at these wavelengths is known as E4/E6, which can characterise the molecular weight and aromaticity of humic acid. A higher value indicates lower aromaticity and smaller relative molecular weight. This ratio is widely used to study the degree of humification in soil organic matter. E4/E6 can be effectively applied to evaluate compost maturity, and a ratio between three and four indicates compost maturity [27,44]. The E4/E6 ratio in the leachate of the microbial nest materials gradually decreased from 7.36 in the initial stage to approximately 1.29 at day 112 (Figure 2g). This decrease may be due to the higher content of macromolecules such as starch, cellulose, proteins, and lignin in the microbial nest materials during the initial stage, resulting in a lower degree of condensation of the aromatic rings in humic acid substances [27]. As fermentation progressed, organic matter was continuously degraded, and humic acid substances gradually increased, leading to a decreased E4/E6 ratio. On day 84, the E4/E6 ratio was only 1.64, indicating that the microbial nest materials tended to mature at approximately 84 days.

### 3.2. Spectral Analysis During the Microbial Nest Pile Fermentation Process

Microbial nest pile (MNP) fermentation involves the transformation of dissolved organic matter (DOM) by microbes. This experiment uses EEM and FTIR to conduct qualitative spectroscopic analysis of the padding materials. Different DOMs contain different functional groups. The analysis of DOM at different fermentation stages using EEM spectroscopy reflects the treatment process and maturity of the MNP. EEM is a spectral plot formed by simultaneously changing the emission wavelength (λEm) and excitation wavelength (λEx), which represent the fluorescence intensity of the measured substance and characteristics of various substances in DOM. These wavelengths were widely applied in the analysis of compost, soil, and other sample types [27,44,45]. The EEM spectrum was divided into five different regions for organic substances: Region I represents aromatic proteins, such as tyrosine (220 nm ≤ λEx ≤ 250 nm, 250 nm ≤ λEm ≤ 330 nm); Region II represents protein-like substances, such as tryptophan and phenylalanine (220 nm ≤ λEx ≤ 250 nm, 330 nm ≤ λEm ≤ 380 nm); Region III represents fulvic acid-like substances (220 nm ≤ λEx ≤ 250 nm, 380 nm ≤ λEm ≤ 500 nm); Region IV represents soluble microbial products (250 nm ≤ λEx ≤ 400 nm, 250 nm ≤ λEm ≤ 380 nm); and Region V represents humic acid-like substances (250 nm ≤ λEx ≤ 400 nm, 380 nm ≤ λEm ≤ 500 nm) [45,46]. The EEM spectra at different MNP fermentation stages (1st, 7th, 28th, 56th, 84th, and 112th day) exhibited significant differences (Figure 3). Protein-like substances and soluble microbial products were dominant during the early stages of MNP fermentation. As fermentation progressed, fulvic acid and humic acid-like substances increased, especially after 84 days of MNP fermentation, when humic acid-like substances became major soluble components. This indicates that the fluorescence intensity of protein-like substances weakens over time, whereas the fluorescence intensity of fulvic acid-like and humic acid-like substances strengthens, in line with changes in the pile material. This demonstrates that the MNP treatment of pig manure is a humification process, and the fluorescence intensity of humic acid-like substances can represent the maturity of the fermentation material.

FTIR spectroscopy is an important technique for studying the organic components of compost leachate and provides information on the molecular structure and functional groups of organic matter [47,48]. To further investigate the composition of soluble substances during MNP fermentation, FTIR spectroscopy was used to analyse the infrared spectra of organic matter in the leachate from different stages of MNP fermentation and characterise the changes in organic matter chemical functional groups during the fermentation process. Appendix A shows the infrared spectra of the leachate samples at different stages of MNP fermentation. The positions of the infrared spectral peaks in the leachate samples at different stages of MNP fermentation did not vary significantly, but the peak intensities showed considerable changes at different stages. There was a prominent absorption peak (peak 1) at 3200–3500 cm^−1^, mainly attributed to the stretching vibration of hydroxyl groups in carbohydrates, such as alcohols, phenols, and organic carboxylic acids, as well as the stretching vibration of N-H bonds in amino acids and amides [47,49]. A weak absorption peak (peak 2) appeared at 2920 cm^−1^, which may have resulted from the stretching vibration of the C-H bonds in the lipids. A strong absorption peak (peak 3) was observed at 1650 cm^−1^, probably from the stretching vibrations of the C=O bonds in amides, carboxylic acids, and ketones, as well as the stretching vibrations of the C=C bonds in aromatic compounds. A strong absorption peak (peak 4) was observed at 1410–1430 cm^−1^, primarily due to the deformation vibrations of -CH_2_ groups connected by double bonds or carbonyl groups in lignin and fatty compounds, as well as the absorption of inorganic NH_4_^+^, NO_3_^−^ and organic carboxylate COO- groups. A strong absorption peak (Peak 5) occurred at 1040–1080 cm^−1^, possibly from the stretching vibrations of C-O bonds in phenols, Si-C bonds in organosilicon compounds, characteristic absorptions of cellulose esters, and stretching vibrations of C-O bonds in polysaccharides. In summary, peaks 1, 2, 3, 4, and 5 were present in the leachate at different stages of MNP fermentation, and their intensities gradually decreased over time. This indicates the continuous degradation of amino acids, alcohols, phenols, amides, ketones, lipids, and aromatic compounds, and it is consistent with the role of the MNP treatment of manure, further confirming that the process is one of humification [50,51,52].

### 3.3. Analysis of Bacterial Community Structure During the MNP Fermentation Process

#### 3.3.1. Diversity of Bacterial Communities

Based on a 97% similarity level, 977,250 high-throughput bacterial reads were obtained from different locations and times in the microbial nest samples. The cluster analysis of these sequences resulted in 1417 OTUs. Alpha diversity was used to analyse the microbial community diversity within a sample. The analysis of diversity within a single sample reflects the richness, diversity, evenness, and coverage of the microbial community. The main alpha diversity indices included the Ace index, Chao1, Shannon, Simpson, Smithwilson, and coverage indices [53,54,55].

The alpha diversity indices at the OTU level are shown in Appendix A. Appendix A show the Ace index and Chao1 indices, respectively. The patterns of these indices were similar, with lower values on the first and seventh days, indicating lower species richness in the samples on these two days. On the first day, manure was not added to the microbial nest as a nitrogen source; therefore, most of the microbial communities could not grow and reproduce. On the seventh day, the low richness could have been due to the high-temperature phase of the pile, where psychrophilic and mesophilic bacteria could not grow or reproduce. Appendix A shows community evenness. Although the samples collected on the first day had lower community diversity, the species within the community were relatively evenly distributed, whereas the samples collected on the 112th day had the opposite distribution. Appendix A show the Shannon and Simpson indices, respectively, which reflect the diversity of the sample community. The Shannon index for the sample on the first day was the lowest, indicating a lower community diversity than that in the other batches (Appendix A). Simpson’s index of the sample on the first day was the highest, indicating a lower community diversity than the other batches (Appendix A). This could be due to the lack of a nitrogen source, which inhibited the growth and reproduction of various microorganisms during the early fermentation stage of the pile. Appendix A shows the species coverage of the samples. The coverage index was above 0.995, indicating that the sequencing results reflected the true condition of microorganisms in each batch of samples.

#### 3.3.2. Bacterial Taxonomy Composition

The composition of the bacterial community at the phylum level during composting at different time points is shown in Figure 4a. From days 1 to 112, the dominant phyla were Proteobacteria, Actinobacteria, Firmicutes, Chloroflexi, Bacteroidetes, Gemmatimonadetes, and Myxococcota. On day 1, Proteobacteria were the most abundant, accounting for over 80% of the community. As the temperature increased, the abundance of Actinobacteria and Firmicutes increased, whereas that of Proteobacteria gradually decreased. This may be attributed to the favourable growth and metabolism of Actinobacteria and Firmicutes at higher temperatures [12,56]. In the later stages of microbial composting, as the temperature decreased, the abundance of Chloroflexi and Bacteroidetes increased. This could be due to a decrease in the efficiency of swine manure decomposition, which led to the proliferation of indigenous bacterial species associated with swine manure in the composting system [4,57].

To further understand the changes in the bacterial community during the microbial composting process in the microbial nest, we conducted a genus-level analysis of the top 50 most abundant bacterial taxa and performed a clustering analysis of the samples and species (Figure 4b). In the early stages of composting, on day 1, the samples showed high abundances of *Pantoea* (42.55%), *Pseudomonas* (6.36%), *Methanobacterium* (14.11%), and *Sphingomonas* (6.78%). At this time, there was a high abundance of anaerobic and acid-tolerant bacteria, which may have been due to the initial limited amount of oxygen in the composting process. As the temperature gradually increased, particularly on day 28, the abundance of thermophilic bacteria, such as *Bacillus* (22.16%), *Ochrobactrum* (5.31%), *Pseudoxanthomonas* (5.36%), *Promicromonospora* (15.89%), and *Streptomyces* (10.05%) increased. In the later stages of composting, during the cooling phase, from days 84 to 112, bacterial diversity decreased. SBR1031 (47.84%), *Anaerolinea* (3.19%), and *Sphaerotheca* (1.76%) were the most abundant on day 112. Furthermore, the cluster analysis of species in the samples revealed that the species compositions of samples on days 3 and 56 were similar, whereas samples on days 7 and 28 were more closely related. The species composition of the samples on days 1 and 112 differed significantly from that of the other samples.

#### 3.3.3. Relationship Between the Bacterial Community and Physicochemical Properties

To analyse the correlation between the bacterial community and variable environmental factors in the microbial composting process in the microbial nest, RDA was performed at the phylum level (Figure 5a). RDA can detect relationships among environmental factors, samples, and bacterial communities or between any two of them. In this study, a correlation analysis was conducted between the top four ranked species at the phylum level and environmental factors. The environmental factors included temperature, pH, moisture content, EC, NH_4_^+^-N, NO_3_^−^-N, total nitrogen (TN), total carbon (TC), and the C/N ratio. RDA1 and RDA2 explained 68.43% of the total variation, and the order of influence of environmental factors on the bacterial community composition was as follows: moisture content > temperature > C/N ratio > pH > EC > TN > TC > E4/E6 > NO_3_^−^-N > NH_4_^+^-N, with moisture content having the greatest impact on the bacterial community. Similar results have also been reported in previous studies [17,50]. In the early stages of composting, the bacterial community structure was positively correlated with the C/N ratio and TC and negatively correlated with the moisture content and temperature, with Proteobacteria being the most dominant. During the middle stage of composting, temperature positively correlated with TC, E4/E6, and NH_4_^+^-N. The temperature had a significant positive correlation with samples from days 7, 28, and 56, which coincided with the dominance of Actinobacteria and Firmicutes during the high-temperature phase. In the later stages of composting, Chloroflexi showed a positive correlation with the moisture content and pH, indicating predominantly anaerobic fermentation, which is consistent with previous findings [3].

To further understand the correlation between the bacterial community composition at the genus level and environmental factors, the top 20 most abundant species at the genus level were selected for the statistical analysis of their correlation with environmental factors using Spearman’s correlation analysis of the microbial composting process in the microbial nest (Figure 5b). The uncategorised order SBR1031 showed a highly significant positive correlation with environmental factors such as pH, TN, NO_3_^−^-N, EC, and moisture content. It exhibited a highly significant negative correlation with the TC, E4/E6, and C/N ratios, as well as a very significant negative correlation with NH_4_^+^-N. Genera such as *Paracoccus*, *Cellulosimicrobium*, *Clostridium*, *Promicromonospora*, *Saccharopolyspora*, *Ochrobactrum*, *Streptomyces*, *Bacillus*, *Sphingobacterium*, *Mycobacterium*, and *Actinomadura* showed significant or even highly significant positive correlations with temperature. This indicates that these genera were more abundant during the high-temperature phase of composting and that most of them were involved in cellulose degradation. Genera such as *Pantoea*, *Microbacterium*, and *Lactobacillus* showed significant positive correlations with the TC and C/N ratio but a significant negative correlation with TN. This suggests a preference for carbon sources over nitrogen sources, corresponding to the early or late stages of composting dominated by anaerobic reactions [58,59]. In conclusion, the efficiency and capacity of swine manure treatment in microbial nests can be improved by changing environmental factors, adjusting the frequency of turning, and adding padding materials. Therefore, analysis of the microbial nest community structure provides a theoretical basis and support for the maintenance and management of microbial nests.

### 3.4. Analysis of Fungal Community Structure During the MNP Fermentation Process

#### 3.4.1. Diversity of Fungal Communities

Based on a 97% similarity level, 1,279,851 high-throughput fungal reads were obtained from different positions and times in the microbial nest samples. The cluster analysis of these sequences identified 337 OTUs. The results of the sample alpha diversity indices based on the OTU level are shown in Appendix A. Appendix A show Ace and Chao1 indices, respectively. The indices were higher on day 1, day 28, and day 56, indicating higher species richness in these samples. The indices for the remaining days were relatively low, and the lowest index was observed on day 84, suggesting lower species richness in this sample. This indicates that fungal species richness is higher in the early and high-temperature phases of microbial nest composting than in the later stage. Compared with the other samples, the sample on day 112 had a lower evenness (Appendix A), indicating a greater difference in the number of different species in this sample. The Shannon index was the lowest for the sample on day 84, indicating that the community diversity in this sample was lower than the other batches (Appendix A). In contrast, Simpson’s index was the highest for the sample on day 84, which also suggests that the community diversity in this sample was lower than that in other batches (Appendix A). The sample collected on day 56 had the highest Shannon index, indicating a higher community diversity in this sample than in the other batches (Appendix A). Similarly, the sample on day 56 had the lowest Simpson’s index, indicating a higher level of community diversity in this sample than in the other batches (Appendix A). In conclusion, the samples from days 3 to 56 exhibited higher diversity, whereas the samples from the early and later stages of microbial nest composting showed lower diversity. Appendix A represents the sample species coverage; the coverage index was above 0.999 for all batches, indicating that the sequencing results accurately reflected the fungal composition in the respective samples.

#### 3.4.2. Fungal Taxonomy Composition

The fungal community composition at the phylum level during microbial nest composting at different time points is shown in Figure 6a. From days 1 to 112, the dominant phyla were Ascomycota, Basidiomycota, and unclassified fungi. During the high-temperature phase of microbial nest composting, from day 3 to day 84, Ascomycota had the highest species richness, suggesting that it was involved in the decomposition of organic matter such as swine manure and straw in the microbial nest, and a significant portion of this was thermophilic fungi. Additionally, some species of the phylum Basidiomycota were present during the composting process, and previous studies have indicated that Basidiomycota is mainly involved in the degradation of lignocellulosic materials.

To further understand the changes in the fungal community during the fermentation of microbial nests, we analysed the top 50 species at the genus level and performed cluster analyses on the samples and species (Figure 6b). *Solicoccozyma* and *Epicoccum* were predominant in the early stages of microbial nest fermentation. During the high-temperature period of pile fermentation, *Dipodascaceae*, *Talaromyces*, *Fusarium*, *Wallemia*, *Aspergillus*, *Trichoderma*, *Thermomyces*, *Penicillium*, and *Geotrichu* were predominant. These fungi are primarily involved in the degradation of lignin, cellulose, and hemicellulose. In the late stages of compost fermentation, *Arthrobotrys* and *Pseudeurotium* were mainly observed.

#### 3.4.3. Relationship Between the Fungal Community and Physicochemical Properties

The relative ranking of the impact of environmental factors on fungal communities was C/N > water content > EC > E4/E6 > temperature > TN > TC > NO_3_^−^-N > NH_4_^+^-N > pH, and the C/N ratio had the greatest impact on the fungal community (Figure 7a). The fungal group structure in the early stage of fermentation positively correlated with the C/N ratio, TC, NH_4_^+^-N, and E4/E6 but negatively correlated with water content and temperature. In the middle stage of fermentation, temperature positively correlated with water content, conductivity, pH, and TN, and the top five species at the order level were mostly concentrated during this period. The fungal community in the late fermentation samples positively correlated with NO_3_^−^-N, EC, pH, TN, and temperature but negatively correlated with the C/N ratio, TC, NH_4_^+^-N, and E4/E6.

To further understand the correlation between the composition of fungal communities at the genus level and environmental factors, statistical analyses were conducted on the top 20 genera in terms of total abundance. Spearman’s correlation analysis was performed on the relationship between the fungal community structure at the genus level during microbial nest compost fermentation and environmental factors (Figure 7b). The genera *Thermomyces* and *Chaetomium* showed significant positive correlations with EC, TN, and NO_3_^−^-N. Additionally, *Thermomyces* exhibited highly significant positive correlations with moisture content and pH. *Aspergillus* and *Wallemia* showed significant positive correlations with temperature, whereas *Penicillium* showed an extremely significant positive correlation with temperature. *Solicoccozyma* and *Epicoccum* were significantly positively correlated with TC, C/N, and E4/E6. In conclusion, different environmental factors exhibited varying correlations with samples and fungal communities at different time periods.

In summary, through the correlation analysis of the microbial nest community structure and environmental factors, measures such as changing the frequency of turning, varying the amount of piggery slurry spraying, and supplementing bedding material can be adopted to prolong the duration of the high-temperature period and increase the relative abundance of advantageous microorganisms during this period, thereby improving the processing efficiency of the microbial nest in a targeted manner. However, there are still shortcomings in the molecular mechanism research of microbial nest treatment for pig manure. The next step will involve utilising omics technologies such as metagenomics, metatranscriptomics, proteomics, and metabolomics to further study and elucidate the genetic, transcriptional, protein, and metabolic levels.

## 4. Conclusions

Piggery slurry solid–liquid mixture treatment involved warming, high-temperature, and cooling phases in an MNS. Physicochemical and spectral analyses confirmed an ongoing humification process within the system. *Pantoea* and *Pseudomonas*, *Bacillus* and *Pseudomonas*, and *Chloroflexi* and *Planctomycetes* dominated the bacterial microorganisms in the early fermentation, high-temperature, and cooling phases, respectively. Additionally, *Solicoccozyma* and *Rhodotorula*; *Talaromyces*, *Fusarium*, and *Arthrobotrys*; and *Arthrobotrys* and *Pseudeurotium* dominated the fungal microorganisms, respectively. Moisture content and the C/N ratio were the key factors influencing bacterial and fungal community structures, respectively. This study lays a theoretical foundation for optimising piggery slurry treatment using an MNS. The MNS treatment capacity and fermentation time can be enhanced by adjusting the frequency of turning the compost, the spraying volume of pig manure and sewage, and replenishing the amount of bedding materials. However, the molecular mechanism still requires further investigation.

## Figures and Tables

**Figure 1 microorganisms-13-00685-f001:**
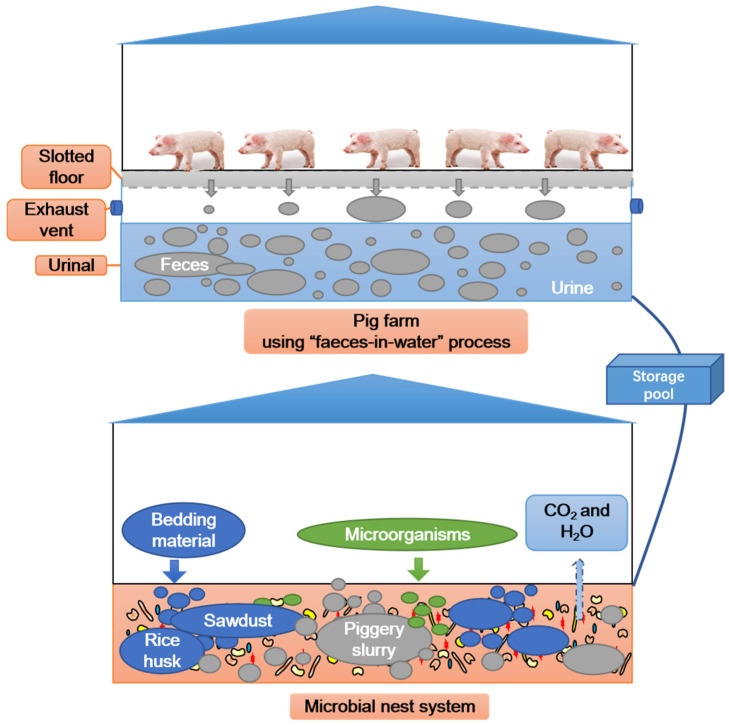
Microbial nest system treatment process flowchart for piggery slurry.

**Figure 2 microorganisms-13-00685-f002:**
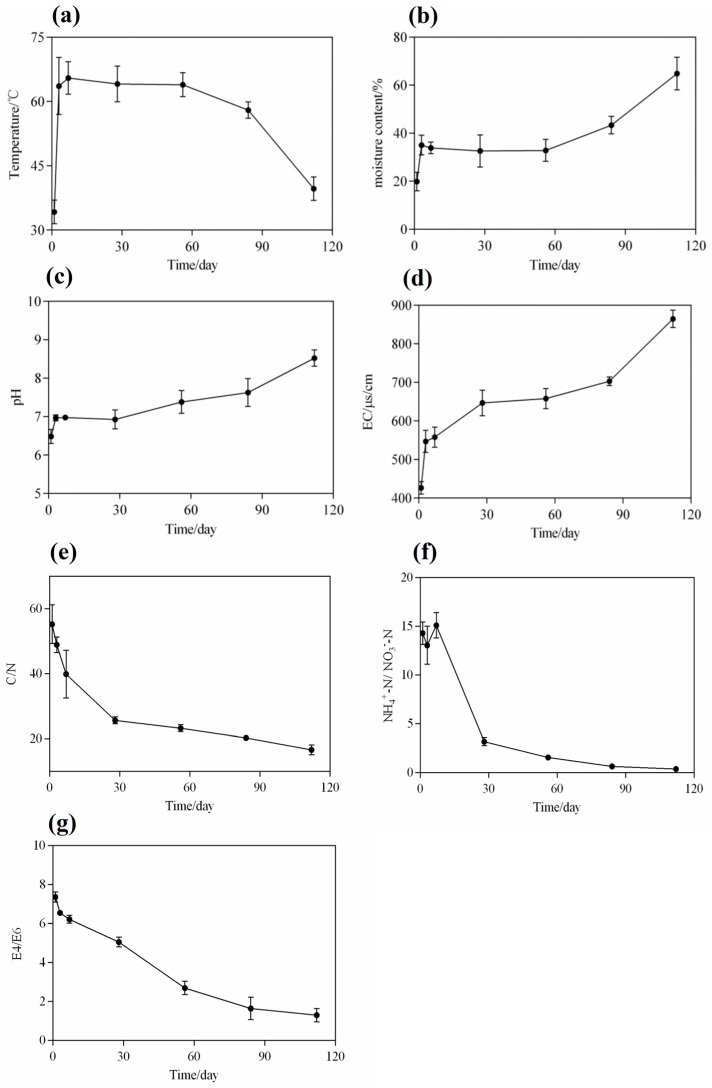
Dynamic changes in the physicochemical properties of compost in the MNS. (**a**) Temperature; (**b**) moisture content; (**c**) pH value; (**d**) electrical conductivity (EC); (**e**) carbon-to-nitrogen ratio; (**f**) ammonium nitrogen-to-nitrate nitrogen ratio; and (**g**) E4/E6. Values are means ± SD (error bars) for three replicates.

**Figure 3 microorganisms-13-00685-f003:**
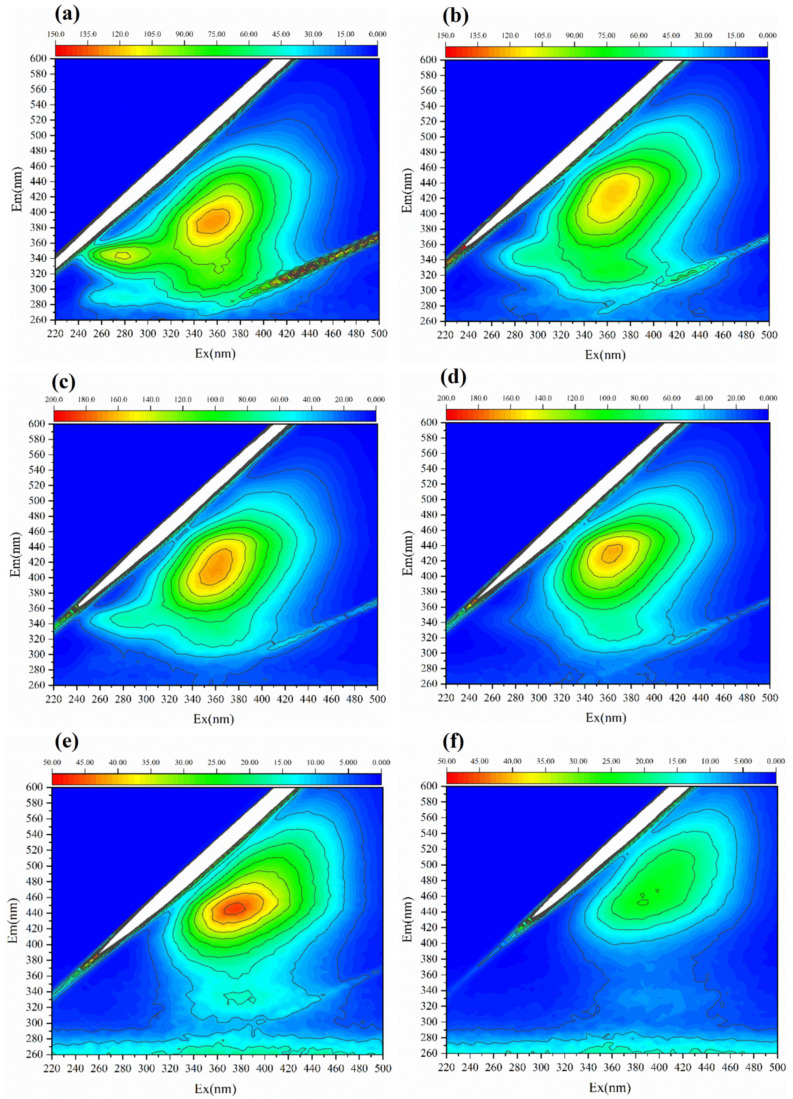
EEM spectroscopy characteristics of different MNP fermentation stages. (**a**) The 1st day; (**b**) 7th day; (**c**) 28th day; (**d**) 56th day; (**e**) 84th day; and (**f**) 112th day.

**Figure 4 microorganisms-13-00685-f004:**
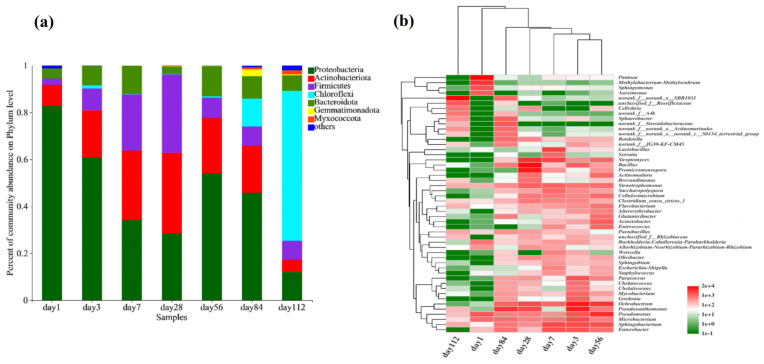
Composition of bacterial communities during fermentation in microbial nests. (**a**) Histogram of bacterial community species composition during microbial nest fermentation at the phylum level; (**b**) heatmap of bacterial community species composition during microbial nest fermentation at the genus level.

**Figure 5 microorganisms-13-00685-f005:**
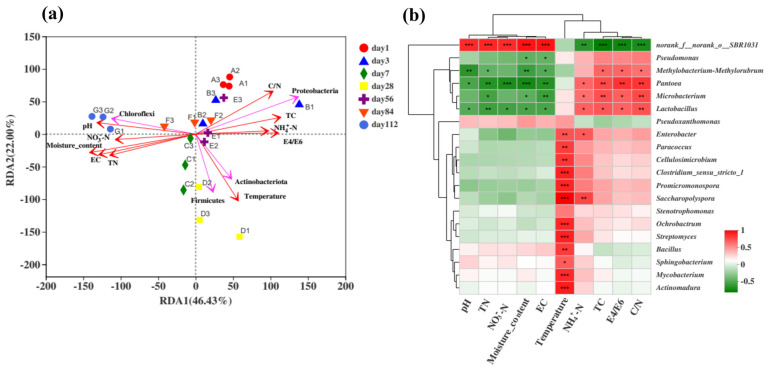
Analysis of the correlation between bacterial communities and environmental factors during the fermentation process of microbial nests. (**a**) The RDA of the correlation between bacterial communities and environmental factors during the fermentation process of microbial nests at the phylum level; (**b**) heatmap analysis of the correlation between bacterial community species composition and environmental factors during the fermentation process of microbial nests at the genus level. Notes: * Correlation is significant at *p* < 0.05. ** Correlation is significant at *p* < 0.01. *** Correlation is significant at *p* < 0.001.

**Figure 6 microorganisms-13-00685-f006:**
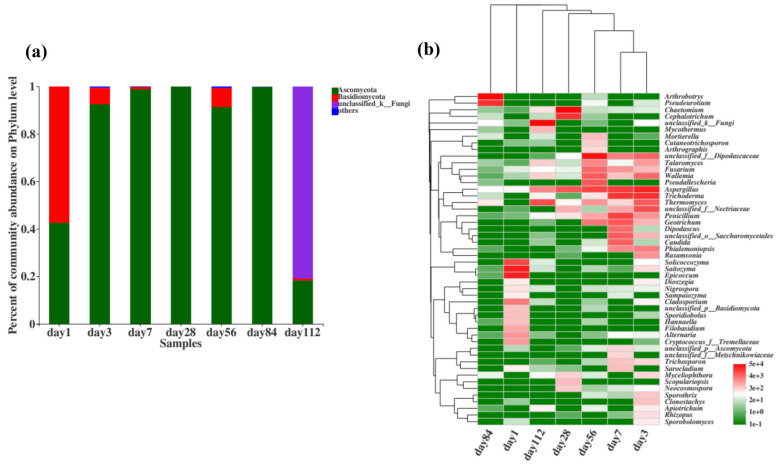
Composition of fungal communities during fermentation in microbial nests. (**a**) Histogram of fungal community species composition during microbial nest fermentation at the phylum level; (**b**) heatmap of fungal community species composition during microbial nest fermentation at the genus level.

**Figure 7 microorganisms-13-00685-f007:**
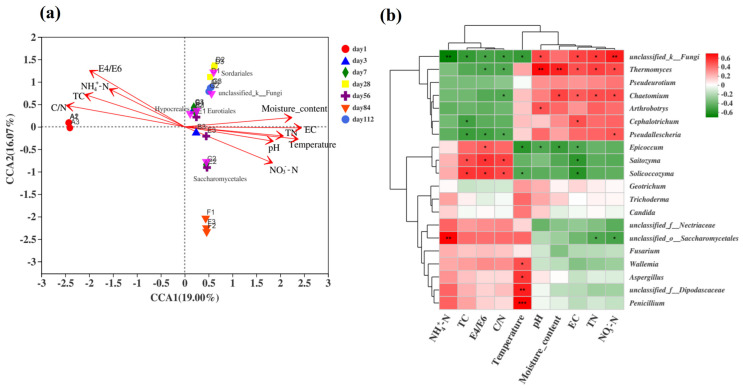
Analysis of the correlation between fungal communities and environmental factors during the fermentation process of microbial nests. (**a**) The RDA of the correlation between fungal communities and environmental factors during the fermentation process of microbial nests at the phylum level; (**b**) heatmap analysis of the correlation between fungal community species composition and environmental factors during the fermentation process of microbial nests at the genus level. Notes: * Correlation is significant at *p* < 0.05. ** Correlation is significant at *p* < 0.01. *** Correlation is significant at *p* < 0.001.

**Table 1 microorganisms-13-00685-t001:** Physical and chemical properties of initial piggery slurry treatment materials.

Material	C/N	pH	Moisture Content (%)	EC (μs/cm)	NH_4_^+^-N/NO_3_^−^-N
Piggery slurry	7.32 ± 1.10	8.80 ± 0.21	92.12 ± 3.12	7520.54 ± 86.26	54.98 ± 6.24
Rice husk	77.66 ± 5.23	5.01 ± 0.13	10.43 ± 1.39	105.65 ± 14.21	1.46 ± 0.98
Sawdust	270.25 ± 10.34	6.21 ± 0.43	7.79 ± 0.32	54.12 ± 4.29	1.24 ± 0.24
Mixed material	55.23 ± 5.94	6.48 ± 0.18	19.87 ± 3.82	426.33 ± 16.29	14.29 ± 1.15

## Data Availability

The data presented in this study are available upon request from the corresponding authors. The data are not publicly available because of company policies.

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
