# Peer review of "Insights into a Novel and Efficient Microbial Nest System for Treating Pig Farm Wastewater"

_microorganisms, 2025, doi:10.3390/microorganisms13030685_

Round 1

Reviewer 1 Report

Comments and Suggestions for Authors

Reviewer Comments

The manuscript presents a study on a novel microbial nest system (MNS) for treating pig farm wastewater, addressing a significant environmental challenge. The research is well-structured and provides valuable insights into the microbial community dynamics and physicochemical changes during the process. The findings contribute to the field of waste treatment and microbial ecology. However, some aspects require minor revisions for clarity, consistency, and completeness.

  1. The abstract could briefly mention the significance of these findings for future applications and concluding sentence summarizing the significance of the findings for practical applications.
  2. Line 53: "However, research on the mechanism of manure and sewage treatment by the MNS has not been reported."
    1. Consider rewording to: "However, the mechanisms underlying manure and sewage treatment in MNS remain poorly understood."
  1. Line 106: "The MNS in this study consisted of three fermentation tanks (45 × 6 × 2 m) sourced from the Bocheng Pig Farm..."
    1. Consider including a simple schematic or diagram to illustrate the experimental setup for better comprehension.
  1. Figures 1-3:
    1. The figures effectively present the data, but the resolution of some images (e.g., spectral analysis graphs) could be improved for better readability.
    2. Ensure all figure captions sufficiently explain trends observed in the data.
  1. Ensure that all figures have consistent formatting (e.g., axis labels, legends).
  1. Line 299: The microbial community analysis results are well-explained.
  1. Consider adding a short discussion on the potential applications of these microbial interactions in optimizing wastewater treatment.
  2. The conclusion effectively summarizes key findings but could benefit from a statement on the broader implications of this research for large-scale implementation.
  1. The authors are suggested to include the following significant references in their manuscript

  • Cui, M. H., Chen, L., Sangeetha, T., Yan, W. M., Zhang, C., Zhang, X. D., ... & Liu, W. Z. (2024). Impact and migration behavior of triclosan on waste-activated sludge anaerobic digestion. Bioresource Technology407, 131094.
  • Zhao, S., Chang, Y., Liu, J., Sangeetha, T., Feng, Y., Liu, D., & Xu, C. (2022). Removal of antibiotic resistance genes and mobile genetic elements in a three-stage pig manure management system: The implications of microbial community structure. Journal of Environmental Management323, 116185.
  • Liu, J., Yu, S., Sangeetha, T., Shi, C., Ju, Y., Yang, C. X., ... & Wang, T. Nutrients Removal and its Relation to the Functional Microbes In Algae-Assisted Sequencing Batch Air-Lift Bioreactor Treating Raw Piggery Wastewater. Available at SSRN 4638488.
  • Niu, S. M., Zhang, Q., Sangeetha, T., Chen, L., Liu, L. Y., Wu, P., ... & Wang, A. J. (2023). Evaluation of the effect of biofilm formation on the reductive transformation of triclosan in cathode-modified electrolytic systems. Science of The Total Environment865, 161308.
  • Wang, L., Liu, C., Sangeetha, T., Yan, W. M., Sun, F., Li, Z., ... & Liu, W. (2023). Integrated microbial electrolysis with high-alkali pretreated sludge digestion: Insight into the effect of voltage on methanogenesis and substrate metabolism. Journal of Environmental Management341, 118007.
  • Gao, L., Sangeetha, T., Wang, L., Cui, M. H., Guo, Z. C., Yan, W. M., ... & Wang, A. J. (2023). The regulating role of applied voltage on methanogenesis in an up-flow single-chamber microbial electrolysis assisted reactor. Journal of Water Process Engineering53, 103799.

Author Response

Comments 1: [ Line 53: "However, research on the mechanism of manure and sewage treatment by the MNS has not been reported."

Consider rewording to: "However, the mechanisms underlying manure and sewage treatment in MNS remain poorly understood."]

Response 1: Agree. Thank you for pointing this out. We have made revisions in accordance with your suggestions.

Revised line 96-98: The sentence “However, research on the mechanism of manure and sewage treatment by the MNS has not been reported.” has been changed to “However, the mechanisms underlying manure and sewage treatment in MNS remain poorly understood.”.

Comments 2: [Line 106: "The MNS in this study consisted of three fermentation tanks (45 × 6 × 2 m) sourced from the Bocheng Pig Farm..."

Consider including a simple schematic or diagram to illustrate the experimental setup for better comprehension.]

Response 2: Agree. Thank you for pointing this out. We have made revisions in accordance with your suggestions.

Figure.1 Microbial nest system treatment process flowchart for pig farm slurry.

Comments 3: [Line 299: The microbial community analysis results are well-explained.

Consider adding a short discussion on the potential applications of these microbial interactions in optimizing wastewater treatment.]

Response 3: Thank you very much for your comments. We have made revisions in accordance with your suggestions.

Revised line 820-826: In summary, through the correlation analysis of microbial nest community structure and environmental factors, measures such as changing the frequency of turning, the amount of piggery slurry spraying, and the supplement of bedding material can be adopted to prolong the duration of the high-temperature period and increase the relative abundance of advantageous microorganisms during the high-temperature period, thereby improving the processing efficiency of the microbial nest in a targeted manner..

Comments 4: [The conclusion effectively summarizes key findings but could benefit from a statement on the broader implications of this research for large-scale implementation.]

Response 4: Thank you very much for your comments. We have made revisions in accordance with your suggestions.

Comments 5: [The authors are suggested to include the following significant references in their manuscript.

Cui, M. H., Chen, L., Sangeetha, T., Yan, W. M., Zhang, C., Zhang, X. D., ... & Liu, W. Z. (2024). Impact and migration behavior of triclosan on waste-activated sludge anaerobic digestion. Bioresource Technology, 407, 131094.

Zhao, S., Chang, Y., Liu, J., Sangeetha, T., Feng, Y., Liu, D., & Xu, C. (2022). Removal of antibiotic resistance genes and mobile genetic elements in a three-stage pig manure management system: The implications of microbial community structure. Journal of Environmental Management, 323, 116185.

Liu, J., Yu, S., Sangeetha, T., Shi, C., Ju, Y., Yang, C. X., ... & Wang, T. Nutrients Removal and its Relation to the Functional Microbes In Algae-Assisted Sequencing Batch Air-Lift Bioreactor Treating Raw Piggery Wastewater. Available at SSRN 4638488.

Niu, S. M., Zhang, Q., Sangeetha, T., Chen, L., Liu, L. Y., Wu, P., ... & Wang, A. J. (2023). Evaluation of the effect of biofilm formation on the reductive transformation of triclosan in cathode-modified electrolytic systems. Science of The Total Environment, 865, 161308.

Wang, L., Liu, C., Sangeetha, T., Yan, W. M., Sun, F., Li, Z., ... & Liu, W. (2023). Integrated microbial electrolysis with high-alkali pretreated sludge digestion: Insight into the effect of voltage on methanogenesis and substrate metabolism. Journal of Environmental Management, 341, 118007.

Gao, L., Sangeetha, T., Wang, L., Cui, M. H., Guo, Z. C., Yan, W. M., ... & Wang, A. J. (2023). The regulating role of applied voltage on methanogenesis in an up-flow single-chamber microbial electrolysis assisted reactor. Journal of Water Process Engineering, 53, 103799.]

Response 5: Thank you very much for your comments. We have inserted the reference into the appropriate position.

Reviewer 2 Report

Comments and Suggestions for Authors

The reviewed manuscript concerns research on the process of treating manure from pig farms using the MNS method. This is a promising technology that allows for the treatment of piggery slurry in a single process, which is a significant improvement of classical methods, in which initially the separation of the liquid phase from the solid phase is used and then each of the phases is treated in separate processes. Therefore, in my opinion, the topic is important from both a scientific and practical point of view.

The research conducted by the Authors was carried out in a way that does not raise any doubts, and adequate analytical and statistical tools were used to analyze the obtained results.

The assumed goals were achieved (the physicochemical and spectroscopic properties of the fermentation pile and its microbiological structure were determined, as well as the impact of selected environmental factors on the process properties).

However, the statement contained in the summary (lines 597-598) that "This study lays a theoretical foundation for optimizing piggery slurry treatment using a MNS" requires better justification. Which of the obtained results are of particular importance in this context and how can they determine the optimum process parameters (e.g. fermentation time, raw piggery slurry load, etc.)

In addition, I would like to make a few detailed comments:

- Part 2.1. - a drawing presenting the scheme of the experimental installation would be a good supplement to the presented description

- Line 123: What was the reason for choosing a specific mixture of bacterial cultures?

- Table 1: What exactly does the term "Mixed material" mean? Is it a mixture of piggery slurry, rice husk and sawdust? If so, how does the moisture content of this material (ca 20%) compare to the value given in line 117 (40-60%)?

- Lines 264-266: Please explain whether the statistical analysis conducted indicates a positive correlation between EC and the content of soluble salts? Or is the presented statement based on other results (e.g. the reference to item 31 of the bibliography)

Author Response

Comments 1: [However, the statement contained in the summary (lines 597-598) that "This study lays a theoretical foundation for optimizing piggery slurry treatment using a MNS" requires better justification. Which of the obtained results are of particular importance in this context and how can they determine the optimum process parameters (e.g. fermentation time, raw piggery slurry load, etc.)]

Response 1: Thank you for your comments

Revised version line 839-844: This study lays a theoretical foundation for optimizing piggery slurry treatment using a MNS. The MNS treatment capacity and fermentation time can be enhanced by adjusting the frequency of turning the compost, the spraying volume of pig manure and sewage, and the replenishment amount of bedding materials. However, the molecular mechanism still requires further investigation.

Comments 2: [- Part 2.1. - a drawing presenting the scheme of the experimental installation would be a good supplement to the presented description]

Response 2: Agree. Thank you for pointing this out. We have supplemented the Microbial nest system treatment process flowchart for piggery slurry.

Comments 3: [Line 123: What was the reason for choosing a specific mixture of bacterial cultures?]

Response 3: Thank you for your comments. The specific mixed microorganisms were selected because they can rapidly initiate the fermentation of the microbial nest system and can raise the temperature of the MNS pile.

Comments 4: [Table 1: What exactly does the term "Mixed material" mean? Is it a mixture of piggery slurry, rice husk and sawdust? If so, how does the moisture content of this material (ca 20%) compare to the value given in line 117 (40-60%)?]

Response 4: Thank you for your comments. "Mixed material" is a mixture of piggery slurry, rice husk and sawdust. A moisture content of 40%-60% is optimal for the microbial nest fermentation process, whereas an initial moisture level of 20% facilitates the easy initiation of fermentation in the microbial nest. These two values are not contradictory.

Comments 5: [ Lines 264-266: Please explain whether the statistical analysis conducted indicates a positive correlation between EC and the content of soluble salts? Or is the presented statement based on other results (e.g. the reference to item 31 of the bibliography)]

Response 5: Thank you for your comments.

The sentence “The EC during the fermentation process of the microbial nest positively correlated with the concentration of soluble salts to some extent” has been changed to “During the microbial nest fermentation process, based on the data shown in Figure 2d, the content of EC is positively correlated with the concentration of soluble salts to a certain extent, as statistically analyzed.”.

Reviewer 3 Report

Comments and Suggestions for Authors

The only suggestion I have for authors is:
Eutrophication is produced by the high levels in N and P that these wastes have not by the high chemical demand of oxygen, if it is true that this can cause the death of aquatic organisms by the consumption of dissolved oxygen.

Author Response

Comments 1: [The only suggestion I have for authors is:

Eutrophication is produced by the high levels in N and P that these wastes have not by the high chemical demand of oxygen, if it is true that this can cause the death of aquatic organisms by the consumption of dissolved oxygen.]

Response 1: Thank you for your comments. 

The sentence “Additionally, the discharge of wastewater with high chemical oxygen demand can cause eutrophication, which could result in the death of aquatic organisms and ecological imbalance” has been changed to “Furthermore, the discharge of wastewater with high nitrogen and phosphorus contents can lead to eutrophication, which may result in the death of aquatic organisms and ecological imbalance.”.
